# Examining allergy related diseases in Africa: A scoping review protocol

**Max Yang Lu** [1], **Nadia Shobnam** [1]*, **Alicia A. Livinski** [2], **Sarini Saksena**[1],
**Dylan Salters**[1‡], **Michelle Biete**[1‡], **Ian A. Myles**[1]

**1** National Institute of Allergy and Infectious Diseases, National Institutes of Health, Bethesda, Maryland, United States of America, **2** National Institutes of Health Library, Office of Research Services, National Institutes of Health, Bethesda, Maryland, United States of America

☯ These authors contributed equally to this work.
‡ DS and MB also contributed equally to this work.
* nadia.shobnam@nih.gov

## Abstract

During recent decades, allergy related diseases have emerged as a growing area of concern in developing regions of the world, including Africa. Worldwide prevalence of allergic diseases has grown to an estimated 262 million for asthma, 400 million for allergic rhinitis (or hay fever), 171 million with atopic dermatitis (or eczema), and over 200 million for food allergy. In Africa, considerable variability exists in the data surrounding prevalence at the continent-wide, regional, and study site levels. Furthermore, research conducted in many rural areas and underdeveloped countries in Africa remains limited, and presently, little has been done to characterize and map the extremely heterogeneous body of literature which confounds research efforts. This scoping review will seek to identify studies examining the prevalence, management strategies, outcomes, and associated risk factors for allergy related diseases in Africa. The Joanna Briggs Institute's scoping review methods will be followed, and the Preferred Reporting Items for Systematic Reviews and Meta-Analyses Extension for Scoping Review (PRISMA-ScR) was used for writing the protocol. Four databases (Embase, Global Health, PubMed, African Journals Online) will be searched for literature published from 2003 to 2023 in any language. Title and abstract screening and full-text screening will be completed by two independent reviewers using Covidence; conflicts resolved by a third reviewer. Data will be extracted using Covidence by two reviewers independently. To report the results, we will follow the PRISMA-ScR checklist and report descriptive statistics and a narrative summary.

## Introduction

Major allergy related diseases (ARDs) include atopic dermatitis (AD) or eczema, asthma, allergic rhinitis (AR) or hay fever, and food allergy. These four diseases are also classified as atopic diseases, a classification of immune disorders commonly associated with the development of the immunoglobulin IgE against typically innocuous allergens [1]. These four conditions have also been linked to one another through shared host and/or environmental factors in a

**Funding:** This work was supported by the Intramural Research Program of the National Institute of Allergy and Infectious Diseases (NIAID) and the National Institutes of Health (NIH). The funders did not and will not have a role in study design, data collection and analysis, decision to publish, or preparation of the manuscript.

**Competing interests:** The authors have declared that no competing interests exist.

phenomenon known as the atopic march [1]. Among established risk factors common between these diseases, prominent ones include diet, airborne exposures, pathogen exposure, industrialization, and climate [2–7].

Atopic dermatitis is an inflammatory skin disease characterized by flares of scaly erythematous rash, predisposition to bacterial superinfection, and severe debilitating pruritus [8, 9]. Though its pathophysiology is multifactorial and has yet to be completely elucidated, it entails a complex interplay between factors including dysfunction of the epidermal barrier, as well as immune and microbiome dysregulation [8, 10].

Asthma, a respiratory disorder, though heterogeneous in nature, is generally accepted to be a chronic inflammatory disorder which may present with clinical symptoms such as wheezing, coughing, and chest tightness [11]. While there are non-allergic phenotypes of asthma [11], allergic asthma, characterized by an association between asthma and environmental allergens, represents the predominant asthma phenotype [12]. It is estimated around 80% of childhood asthma and 50% of adulthood asthma has an allergic component [13].

Allergic rhinitis, is an inflammatory disease of the nasal mucosa triggered by allergens [14]. Common symptoms include nasal congestion, rhinorrhea (runny nose), sneezing, and nasal pruritus (itching) [15]. When ocular symptoms manifest, the disorder is also termed allergic rhinoconjunctivitis (ARC) [15].

Food allergy is the consequence of an immune system reaction caused by exposure to a food protein antigen, sometimes in very small quantities [16]. This disorder can manifest a wide-variety of symptoms ranging from mild to life-threatening [16]. The immune system reactions responsible for food allergy may be IgE mediated, mediated by both IgE-dependent and -independent pathways (mixed), or be non-IgE-mediated [16].

These ARDs represent a significant burden for public health worldwide and are not only detrimental to individuals' own quality of life, but also impose socioeconomic costs on both individuals and societies [17]. One report from 2019 estimated 262 million cases of Asthma globally creating a disease burden 21.6 disability adjusted life years (DALYs) with another 171 million atopic dermatitis cases responsible for 7.48 million DALYs [18]. Additionally, the global prevalence of food allergy and allergic rhinitis has been estimated at 200–250 million and 400 million respectively [19]. For asthma, while the highest prevalence exists in developed countries, death rates are highest in low and lower middle-income countries, classifications possessed by many countries in Africa [20]. Furthermore, globally, African countries, especially those in Sub-Saharan Africa, possess some of the highest disease burdens of asthma as measured via DALYs [18]. These unfavorable outcomes arise in Africa due to a variety of challenges surrounding disease management related to diagnosis and lack of access to treatment and care. One study which assessed the availability of three asthma drugs on the World Health Organization Model List of Essential Medicines (beclomethasone, budesonide, and salbutamol) in 24 African countries demonstrated limited availability of these drugs [21]. Beclomethasone and budesonide were unavailable across all health facilities (private pharmacy, national procurement center, and public hospital pharmacy) and across all forms (reference brand, innovator brand, or generic product) in eleven and sixteen African countries surveyed respectively [21]. Even when available in a country, such availability was often limited to a certain form or to a certain class of health facility [21]. Another review article which examined existing guidelines for Atopic Dermatitis and Asthma management across 25 African countries found national guidelines to be either entirely nonexistent or highly variable between countries, suggested to be a consequence of the wide range of economic conditions between countries [22]. This poor healthcare infrastructure contributes to outcomes such as under- or misdiagnosis as highlighted by studies of asthma from Senegal, Nigeria, and Uganda [22]. Regarding food allergy in Africa, epidemiological data to inform policy remains scarce, and

poor economic conditions and healthcare infrastructure further exacerbate management challenges [23].

While there are existing reviews examining prevalence and management of major ARDs across Africa [22–34], only a handful of studies were systematic reviews and employed a standardized and rigorous methodology to review the existing literature [25, 27, 30]. These systematic reviews also only examined asthma and atopic dermatitis and were published from 2010–2013. Additionally, many multi-site studies, review articles, and systematic analyses suggest the prevalence of these four diseases is evolving in Africa [23, 25, 28, 30, 33, 35–38], thus, revisiting the matter is warranted in order to examine and map the most current literature. Furthermore, among studies examining trends in prevalence, only a limited number of study sites have been employed and there remains significant variability in conclusions surrounding trends of prevalence at not only the continent-wide level, but also between study sites and regions. There also exists a scarcity of reviews with focuses on other aspects of population health surrounding these diseases in Africa such as associated risk factors or outcomes. To the best of our knowledge, there has been no recent endeavor to perform a scoping review on existing literature studying ARDs in the continent of Africa. There remains a need to synthesize such evidence, especially given atopic march and the associations between these conditions with one another. We aim to conduct a scoping review examining the literature published from 2003 to 2023 to map the most current research on ARDs (atopic dermatitis, asthma, allergic rhinitis, and food allergy) in Africa, in particular examining prevalence across settings, management strategies, associated risk factors, and disease outcomes. By highlighting gaps in research surrounding ARDs across Africa, including those regarding study settings, the scoping review will aim to inform future research and policy.

## Methods

The proposed scoping review will follow the Joanna Briggs Institute (JBI) scoping review methods [39]. The results will be presented in accordance with the Preferred Reporting Items for Systematic Reviews and Meta-Analysis: Extension for Scoping Reviews (PRISMA-ScR) [40].

### Research question

The main research question is: What is known about allergic diseases (asthma, atopic dermatitis (or eczema), food allergy, and allergic rhinitis (or hay fever or allergic rhinoconjunctivitis)) in Africa?

The research sub-questions are:

a. What is the prevalence of these allergy related diseases in Africa?

b. What settings (country, urban vs. rural) are these studies being published from in Africa?

c. What demographic and environmental risk factors exist for these allergic diseases in Africa?

d. What is the nature, extent, and efficacy of current management strategies for these allergy related diseases in Africa?

e. What are the outcomes for those afflicted by these allergy related diseases in Africa?

### Eligibility criteria

Table 1 includes the eligibility criteria that will be used for study selection. For the purposes of this review, studies on the Canary Islands will not be included, as although they are

**Table 1. Inclusion and exclusion criteria.**

| | |
|---|---|
| **Inclusion Criteria** | • Must be focused on population health of the continent of Africa (outside of the Canary Islands), not on populations of African ancestry outside of the continent<br>• Must have a main focus on or the principal findings must concern at least one of the four allergy related diseases<br>• Must include a focus on prevalence of allergy related diseases, associated demographic or environmental risk factors, management strategies, or outcomes<br>• Any language<br>• All ages<br>• Human-based research |
| **Exclusion Criteria** | • Lack of original data or original statistical analysis (study protocols, narrative reviews, book chapters, systematic reviews etc.)<br>• Prospective modeling studies and meta-analyses<br>• Primary focus does not surround allergy related disease<br>• Lack of focus on population health (studies primarily in a laboratory setting, case-reports)<br>• Not conducted in Africa (besides the Canary Islands) or for multi-center studies across continents, data from the African site(s) alone is not extractable<br>• Ethnomedicinal studies<br>• Allergic diseases solely evaluated as risk factors or comorbidities |

geographically African, as a part of Spain, this region remains demographically, politically, and economically distinct. Management strategies will encompass studies examining assessment and diagnostic tools, interventions, treatment adherence, and those surrounding disease stakeholder knowledge and attitudes.

For a study to be defined as possessing a primary focus on allergic diseases either: (1) the primary stated objective must involve one of the four allergic diseases or (2) the primary findings of the study (the findings when investigating the primary stated objective) must surround at least one allergic disease. Additionally for a study to be defined as possessing a primary focus, data regarding allergic diseases must also be independently presented and extractable independent of other non-allergic diseases. The one notable exception to this definition of a primary focus is that studies in which allergic diseases are solely evaluated as risk factors or comorbidities will be excluded, even when this is the primary study aim or finding. This exception was developed for the purpose of managing the scope of the review and promoting clarity and reproducibility of the inclusion and exclusion criteria during screening.

Laboratory setting studies include but are not limited to in vitro based studies and pharmacological studies. Outcomes refer not only to clinical outcomes, but also the disease burden of patients and stakeholders. Only human-based studies will be included, excluding studies such as those evaluating pollen allergy counts.

## Information sources and search

The search strategy was developed by a research librarian (AAL) and guided by the Population-Concept-Context (PCC) framework shown in Table 2. The following databases will be

**Table 2. PCC framework.**

| | |
|---|---|
| **Population** | • Human based research<br>• All ages |
| **Concept** | • Allergic diseases (asthma, atopic dermatitis (or eczema), food allergy, and allergic rhinitis (or hay fever or allergic rhinoconjunctivitis))<br>• Population health (prevalence, management, outcomes, environmental and demographic risk factors) |
| **Context** | • Studies on the African continent<br>• Studies from 2003–2023 |

searched: Embase (Elsevier), Global Health (CABI), PubMed (US National Library of Medicine), and African Journals Online. We will exclude the grey literature. The searches will be limited to those published in any language from 2003 to 2023. The time frame of the search will be limited to the last two decades as to include the most recent and pertinent literature, since many reports suggest evolving prevalence of these diseases [23, 25, 28, 30, 33, 35–38]. Furthermore, a preliminary literature search was conducted and uncovered a relatively low number of publications before 2003. The exact search terms will be adapted for each database, and the search strategy to be used for PubMed is included in S1 Appendix. Additionally, reference lists of included articles and appropriate review articles will also be scanned by a minimum of one reviewer to identify other potentially relevant articles. Any articles identified through these supplemental methods will proceed through the study selection process outlined below.

EndNote 20 (Clarivate Analytics) will be used to collect, manage, and identify duplicate citations from the database searches. The unique articles identified will be exported into Covidence (Veritas Health Innovations, Ltd) for study selection.

## Selection of sources of evidence

Each article will be screened according to the inclusion and exclusion criteria (Table 1) developed following the PCC framework (Table 2). The inclusion and exclusion criteria will be iteratively refined during title and abstract screening to confirm their clarity and reproducibility. To reduce the risk of bias and errors, each article will be screened by two independent reviewers first at the level of title and abstract, and then subsequently the full text. During title and abstract screening, disagreements will be resolved by the corresponding author (NS). During full-text screening, conflicts will be resolved via group discussion or a third independent reviewer. Prior to commencing study selection, a pilot of 40 records randomly selected by the librarian will be conducted at both levels with all reviewers in Covidence. After the pilot, the eligibility criteria will be further refined.

## Data charting process and data items

After full-text screening the data extraction process will be piloted with all reviewers using 5 articles. During the pilot, modifications will be made to the data collection process and data items as necessary and documented in the protocol. For data collection, the relevant data from each study will be extracted by two independent reviewers using Covidence. Any discrepancies in the data will be reviewed and resolved by a separate third reviewer, and if necessary, consensus discussion with the two reviewers. For multi-center studies including sites outside of the African continent, only data from the African study sites will be extracted; if that is not feasible the article will be excluded.

The proposed data items to be extracted for the summary data table are:

1. Article title

2. Lead author last name

3. Publication year

4. Journal name

5. Study setting (Country and city, urban vs rural vs mixed vs semi-urban as reported)

6. Participant sex (male only, female only, both sexes, not reported)

7. Age (as reported)

8. Clinical characteristics of participants (whether the population was a representative subset or if it was a group that had the allergic disease)

9. Final total sample size (or for each study site if there are multiple)

10. Study methods and design (cross-sectional, prospective cohort, retrospective cohort, qualitative, case-control, randomized control trial, meta-analysis)

11. Study objective/aims

12. Demographic or environmental risk factors associated with allergic disease prevalence if mentioned

13. Demographic and environmental risk factors associated with allergic disease severity if mentioned

14. Demographic and environmental risk factors associated with extent of allergic disease control and management (including barriers to management) if mentioned

15. Lifetime and/or current prevalence of allergy related disease if mentioned

16. How the presence of the disease was assessed/defined if mentioned

17. Aspect of allergy related disease management assessed if mentioned (stakeholder education, care-seeking behavior, accessibility of care, medication, public policy, assessment and diagnostic tools)

18. Which allergy related disease(s) was studied (Including specific food allergen(s) if food allergy related)?

19. If studied, among what groups of stakeholders were knowledge and attitudes assessed (physicians, nurses, other providers, patients, familial caregivers)

20. Type of outcome assessed (socioeconomic burden, risk for comorbid disease, severity of symptoms, mortality)

21. Other key findings relevant to research questions

## Analysis and synthesis of results

The collected data will be cleaned using Microsoft Excel. The data extracted will be tabulated and descriptive statistics analyzed using Excel. Frequency counts of various aspects of included articles will be tabulated such as setting (country, urban vs rural), study design, sample age (pediatric vs adult), aspect of population health described (management vs prevalence vs outcomes vs associated risk factors), type of allergic disease, types of associated risk factors, disease assessment methods, and types of management strategies employed. Frequency counts may also be analyzed at multiple levels (i.e., frequency counts by country for studies on asthma). Such results will be presented in tabular form and potentially maps for analysis of study setting. A narrative summary will accompany and describe the tabular and graphical results to highlight the findings in relation to the principal research question and sub-questions. The narrative summary will also highlight research gaps and implications for future research.

## Ethics and dissemination

Ethical approval is not required as the study is based on existing data. We plan to disseminate the final report via a peer-reviewed journal and potentially relevant conferences as well. Any

potential amendments made to this protocol will be discussed in the final report, as well as the rationale for such changes.

## Discussion

The aim of our work is to gather and map the existing body of literature surrounding allergic diseases in Africa, specifically examining aspects of population health. We will conduct a scoping review employing a comprehensive literature search and systematic article screening and data extraction. Mapping the evidence surrounding prevalence, associated risk factors, management, and outcomes of the diseases will serve to inform future research and highlight current gaps in the body of literature.

Employing the methodology of a scoping review will serve to reduce bias and capture a wide variety of studies encompassing many methodologies and study designs to answer our broad research question. The inclusion of studies in all languages further serves to promote the breadth of research captured. However, despite the carefully curated search query with the involvement of a librarian and independent screening of articles by two reviewers, there remains the possibility that certain relevant studies may be excluded. Similarly, for the purpose of feasibility and managing the scope of the review, studies in which the main findings did not concern allergic diseases will be excluded. Additionally, despite the strengths of the scoping review methodology, it presents limitations regarding the fact that studies are not typically assessed for quality. For example, operational definitions of asthma are often inconsistent between different studies [41]. While the review will provide an overview of the existing literature, such limitations may affect the strength of the conclusions able to be drawn.

However, despite the limitations, to the best of our knowledge our scoping review will represent the first such endeavor examining allergic diseases in Africa. Any changes to the study protocol will be reported in the final scoping review, which we aim to disseminate via a peer-reviewed journal as well as relevant conferences.

## Supporting information

**S1 Checklist. Preferred Reporting Items for Systematic reviews and Meta-Analyses extension for Scoping Reviews (PRISMA-ScR) checklist.**
(DOCX)

**S1 Appendix. Full search strategy for PubMed.**
(DOCX)

## Author Contributions

**Conceptualization:** Nadia Shobnam.

**Methodology:** Max Yang Lu, Nadia Shobnam, Alicia A. Livinski, Sarini Saksena, Dylan Salters, Michelle Biete.

**Project administration:** Nadia Shobnam, Ian A. Myles.

**Supervision:** Nadia Shobnam, Ian A. Myles.

**Validation:** Max Yang Lu, Nadia Shobnam, Alicia A. Livinski, Sarini Saksena, Dylan Salters, Michelle Biete.

**Writing – original draft:** Max Yang Lu.

**Writing – review & editing:** Max Yang Lu, Nadia Shobnam, Alicia A. Livinski, Ian A. Myles.

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
