## [Decision Letter · Decision Letter 0]

2 Nov 2023

PONE-D-23-28766Examining Allergy Related Diseases in Africa:  A scoping review protocolPLOS ONE

Dear Dr. Shobnam,

Thank you for submitting your manuscript to PLOS ONE. After careful consideration, we feel that it has merit but does not fully meet PLOS ONE’s publication criteria as it currently stands. Therefore, we invite you to submit a revised version of the manuscript that addresses the points raised during the review process.

We look forward to receiving your revised manuscript.

Kind regards,

Pisirai Ndarukwa, Ph.D.

Academic Editor

PLOS ONE

Journal Requirements:

Reviewers' comments:

Reviewer's Responses to Questions

**Comments to the Author**

1. Does the manuscript provide a valid rationale for the proposed study, with clearly identified and justified research questions?

Reviewer #1: Yes

2. Is the protocol technically sound and planned in a manner that will lead to a meaningful outcome and allow testing the stated hypotheses?

Reviewer #1: Yes

3. Is the methodology feasible and described in sufficient detail to allow the work to be replicable?

Reviewer #1: Yes

4. Have the authors described where all data underlying the findings will be made available when the study is complete?

Reviewer #1: Yes

5. Is the manuscript presented in an intelligible fashion and written in standard English?

Reviewer #1: Yes

6. Review Comments to the Author

You may also provide optional suggestions and comments to authors that they might find helpful in planning their study.

Reviewer #1: Thank you for the opportunity to review this article/study protocol. It is interesting and I look forward to reading the findings of the actual study. I have a few minor concerns/clarifications.

1. In some aspects, the writing style appears like the review has already been conducted. For instance, line 123 “Table 1 includes the eligibility criteria used for study selection.”; line 148 “The time frame of the search was limited to the last two decades as to include the most recent and pertinent literature”; line 237 “Similarly, for the purpose of feasibility and managing the scope of the review, studies in which the main findings did not concern allergic diseases were excluded”; etc.

2. I see this statement “The searches will be limited to those published in any language from 2003 to 2023” in line 147-149 to be a repetition of what is already stated in the inclusion/exclusion criteria

3. Line 163-164:” To reduce the risk of bias and errors, each article will be screened by two independent reviewers (MYL, NS, SS, DS, MB)” Do the authors mean four independent reviewers? Because the names in the parentheses are more than two. Or they will do it in pairs?

4. Authors indicate that “During both levels of screening, disagreements will be resolved by the corresponding author (NS)”, line 165. It appears, however, that the correspondent author (NS) is part of the independent reviewers stated in line 164. How then does he/she resolve a disagreement in an unbiased manner?

5. A piece of advice and authors may ignore/disregard it. Assessment of methodological quality of reviewed studies, in most instance, is not part of scoping reviews, but adding this to the review will help improve the overall quality of the study.

7. PLOS authors have the option to publish the peer review history of their article (what does this mean?). If published, this will include your full peer review and any attached files.

Reviewer #1: No

---

## [Author Response · Author response to Decision Letter 0]

6 Nov 2023

To: Pisirai Ndarukwa

Academic Editor, PLoS ONE

Dear Dr. Ndarukwa,

Thank you for inviting us to submit a revised version of our manuscript Examining allergy related diseases in Africa: A scoping review protocol to PLoS ONE. We also thank you for the time and effort both you and the reviewer have put into providing feedback to improve our paper. We have carefully considered the points raised and have made a faithful effort to improve our paper.

To facilitate your review of our revisions, below is a point-by-point response to the concerns/clarification points raised:

1. In some aspects, the writing style appears like the review has already been conducted. For instance, line 123 “Table 1 includes the eligibility criteria used for study selection.”; line 148 “The time frame of the search was limited to the last two decades as to include the most recent and pertinent literature”; line 237 “Similarly, for the purpose of feasibility and managing the scope of the review, studies in which the main findings did not concern allergic diseases were excluded”; etc.

Thank you for pointing this out. In these specified instances, we have revised the tense of the writing from past to future. Additionally, in another similar situation in line 243 of the version without track changes (“our scoping review will represent”), the tense has also been revised from past to future.

2. I see this statement “The searches will be limited to those published in any language from 2003 to 2023” in line 147-149 to be a repetition of what is already stated in the inclusion/exclusion criteria

We agree that this is repetitive. To most accurately describe our protocol while eliminating this repetition, we have deleted “Published from 2003-2023” from Table 1 regarding the inclusion/exclusion to most accurately describe our methodology. We have opted to remove this description from the inclusion/exclusion criteria table rather than the search strategy section as we will not be voting to include or exclude articles based on publication date during screening. Instead, the search strategy will have effected such a time-restriction.

3. Line 163-164:” To reduce the risk of bias and errors, each article will be screened by two independent reviewers (MYL, NS, SS, DS, MB)” Do the authors mean four independent reviewers? Because the names in the parentheses are more than two. Or they will do it in pairs?

We agree that this phrasing can be confusing. We have deleted the initials of the screeners in lines 163-164 and also when describing the data extraction process in line 174 in order to promote clarity, as well as to reflect the fact that additional researchers who are not co-authors on this protocol may aid in screening and data extraction in the final scoping review. Additionally, in line 174, the phrasing has been revised to “by two independent reviewers” to clarify that each study will only be reviewed by two reviewers total.

Similarly, to promote accuracy and clarity regarding our methodology of conflict resolution in data extraction and charting, we have changed lines 175-177 of the track changes edition to read “Any discrepancies in the data will be reviewed and resolved by a separate third reviewer, and if necessary, consensus discussion with the two reviewers.”

4. Authors indicate that “During both levels of screening, disagreements will be resolved by the corresponding author (NS)”, line 165. It appears, however, that the correspondent author (NS) is part of the independent reviewers stated in line 164. How then does he/she resolve a disagreement in an unbiased manner?

Thank you for pointing this out. We agree that the corresponding author cannot resolve a disagreement in an unbiased manner if she is one of the independent reviewers who voted on the study in the first place. Thus, we have amended our criteria surrounding conflict resolution. The corresponding author (NS) will still resolve disagreements at the level of title/abstract screening to facilitate ease and feasibility of conflict resolution. However, at the level of full-text screening, disagreements will be resolved via group discussion or an independent, separate third reviewer. This ensures that at the final and most thorough stage of screening to determine inclusion or exclusion of articles, each article will undergo a rigorous and unbiased review including during potential conflict resolutions. 

5. A piece of advice and authors may ignore/disregard it. Assessment of methodological quality of reviewed studies, in most instance, is not part of scoping reviews, but adding this to the review will help improve the overall quality of the study. 

Thank you for this suggestion. We considered conducting assessment of study quality as part of our review, however, given that it ultimately is not an element of most scoping reviews, decided against such a step. We also have already taken some other measures towards ensuring a high quality of included studies such as excluding grey literature.

Thank you again for providing the opportunity for us to strengthen our manuscript and we hope that it is now suitable for publication in PLoS ONE. We look forward to hearing from you.

Sincerely,

Nadia Shobnam

National Institute of Allergy and Infectious Diseases (NIAID)

Email: nadia.shobnam@nih.gov

Phone: 302-345-1118

---

## [Decision Letter · Decision Letter 1]

16 Jan 2024

Examining Allergy Related Diseases in Africa:  A scoping review protocol

PONE-D-23-28766R1

Dear Dr. Shobnam,

We’re pleased to inform you that your manuscript has been judged scientifically suitable for publication and will be formally accepted for publication once it meets all outstanding technical requirements.

Kind regards,

Pisirai Ndarukwa, Ph.D.

Academic Editor

PLOS ONE

Additional Editor Comments (optional):

Reviewers' comments:

Reviewer's Responses to Questions

**Comments to the Author**

1. Does the manuscript provide a valid rationale for the proposed study, with clearly identified and justified research questions?

Reviewer #1: Yes

2. Is the protocol technically sound and planned in a manner that will lead to a meaningful outcome and allow testing the stated hypotheses?

Reviewer #1: Yes

3. Is the methodology feasible and described in sufficient detail to allow the work to be replicable?

Reviewer #1: Yes

4. Have the authors described where all data underlying the findings will be made available when the study is complete?

Reviewer #1: Yes

5. Is the manuscript presented in an intelligible fashion and written in standard English?

Reviewer #1: Yes

6. Review Comments to the Author

You may also provide optional suggestions and comments to authors that they might find helpful in planning their study.

Reviewer #1: Thank you for the opportunity to review the revised version of this manuscript. The authors have addressed/clarified all the issues I raised. Therefore, I have no further comments.

Thank you

7. PLOS authors have the option to publish the peer review history of their article (what does this mean?). If published, this will include your full peer review and any attached files.

Reviewer #1: No

---

## [Editor Report · Acceptance letter]

10 Feb 2024

PONE-D-23-28766R1 

PLOS ONE

Dear Dr. Shobnam, 

I'm pleased to inform you that your manuscript has been deemed suitable for publication in PLOS ONE. Congratulations! Your manuscript is now being handed over to our production team.

Kind regards, 

on behalf of

Prof Pisirai Ndarukwa 

Academic Editor

PLOS ONE